# A Study on the Application of Kiosk Service as the Workplace Flexibility: The Determinants of Expanded Technology Adoption and Trust of Quick Service Restaurant Customers

**Kyung Hwa Seo**

Department of Hotel Culinary Arts and Bakery, Ulsan College, Ulsan 44022, Korea; khseo@uc.ac.kr

**Abstract:** This study presents fundamental data on the technology acceptance of kiosks in QSR (Quick Service Restaurants) and the marketing plans for efficient management performance. In this paper, the combined concept model was established through the acceptance literature, and an expanded UTAUT2 (Unified Theory of Acceptance and Use of Technology) model was then presented by empirical analysis. This study was evaluated by 303 customers with experience using QSR kiosks. The results show that the seven factors suggested by the UTAUT2 and trust have a positive direct and indirect effect on behavioral intention. In addition, this study confirmed the significant influence relationship between the variables in UTAUT2. It was also found that facilitating conditions and price value have a significant positive effect on trust. Lastly, trust has a significant positive effect on performance expectancy and behavioral intentions. As a result, this research demonstrates an extended and integrated UTAUT2 by verifying the relationship between basic UTAUT2 and trust. The limitations of this study and recommendations for future research are also discussed.

**Keywords:** kiosk; technology adoption; QSR; UTAUT2; trust; behavioral intention

## 1. Introduction

The quick-service restaurant (QSR) industry plays an important role in driving the economy and culture around the world. However, the QSR has a particularly difficult problem when creating customer relationships because it is difficult to develop differentiated identities due to a low level of consumer brand loyalty and its standardized products [1]. Nevertheless, companies must build a strong customer-focused relationship marketing strategy to implement their business strategy in a fiercely competitive market environment. In this regard, the existing approach focuses on marketing strategies for customer satisfaction. Until now, studies have not actively reviewed the association with customer behavioral intentions based on a QSR's information communication technology (ICT).

Today's rapidly changing world is accelerating the evolution of innovative technologies by adding information processing capabilities, such as self-service technology (SST), in addition to just information delivery. The hospitality industry is expanding to an ongoing evolving technology adoption process, and new possibilities for technology adoption are constantly being explored [2]. A kiosk is one of the most common types of SST, and in the restaurant industry, it is now recognized as one of the latest innovations [3,4]. The kiosk in the restaurant is an unmanned payment system that allows customers to order food directly or participate in its service processes, rather than experience face-to-face service with employees. Since kiosks in a restaurant not only build an innovative technology infrastructure, but also provide new value for customers, it is expected to become a universal and routinized service for customers within a short time. So, we need to understand the role and importance of kiosks for various marketing activities. In particular, restaurants, such as QSR, offer lower traditional services

by delivering lower selling prices than other types of restaurants, while services, such as speed and efficiency, still remain important even with the limited services [5]. The introduction of Kiosk technology can be an alternative to QSR's service failure, thereby allowing managers to review its positive adoption. Additionally, technological innovation can provide a variety of service experiences to customers while also increasing employee work efficiency [6]. Indeed, building a system by introducing technology can provide positive benefits to both customers and the restaurants [7]. As a result, if you do not understand the relationship that arises between customers and technology in an offline environment, it is difficult to create a specific strategy method for restaurant management, and management can neither be realized for future technological innovation nor business performance successfully.

The purchasing behavior is one of the strong indicators of the supplier–consumer relationship. Regarding the debate presented earlier, ICT is an important key strategy for successful business activities, so it is thus very important to identify significant predictors that relate to the technology acceptance behavior of users for new technologies. Research on consumer behavior as technological advancement has been developed into a Unified Theory of Acceptance and Use of Technology (UTAUT, UTAUT2) using the Technology Acceptance Model (TAM) [8–10]. It can be described as a framework that is best suited for establishing academic development and a theoretical background while verifying the causal relationship with the various variables involved in the consumer's acceptance of new technology. Currently, the developed UTAUT2 plays an important role in enhancing the understanding of consumer behavioral intention. Recently, research on the acceptance of technology by restaurant customers has focused on mobile apps [11–13]. However, the kiosk in QSR differs significantly from online technology acceptance, considering that interactions with customers will occur offline rather than online. In particular, studies related to the kiosk technology acceptance model in the restaurant field be limited [14,15]. For all these reasons, it is necessary to research the topic from various perspectives regarding technology adoption for QSR. Above all, based on the UTAUT2 model, it is necessary to comprehensively review various variables related to the customer's behavioral intention of kiosk technology acceptance. Results will clearly show the important role of kiosks so that managers can suggest specific marketing strategy methods from a customer perspective.

Furthermore, this study additionally further proposes a theoretical framework for the competitive advantage of restaurants. It confirms the importance of trust within the UTAUT2 model. Customers ask themselves questions before using the new system; 'Do I trust the system?'. Since unmanned payment systems, such as kiosk services, have various anxiety factors related to technical defects and personal information leakage, users always raise belief questions when using the system. People generally use trust and familiarity as their primary mechanisms to reduce social uncertainty whenever rules and conventions are not enough to convince them [16]. Indeed, Kim et al. [17] emphasize that trust is an important strategy when dealing with an uncertain and uncontrollable future. In terms of technology adoption, trust has been identified as an important aspect in previous studies [18–20]. A kiosk creates interrelationships between people and technology, and thus trust is paramount in such interrelationships. In the UTAUT2 study, trust was further approached to understand only the direct impact relationship on behavioral intentions [13,21–24]. Still, we do not fully understand a customer's acceptance of technology because a comprehensive review of the UTAUT2 including trust has not yet been conducted.

This study aims to provide fundamental data on Kiosk's marketing strategy plan leading to customer behavioral intentions to QSR's manager. There is a need to first identify the independent factors involved in customers' behavioral intentions based on the UTAUT2 model. In addition, it reviews the notion that perceived trust of kiosks is a key factor for determining customer technology acceptance. By integrating the UTAUT2 model and trust, this study presents an extended UTAUT2 model of QSR. This extended model offers guidance to managers for creating the most effective strategies to gather new customers and retain loyal customers, and these actions will help with the successful performance of QSR. Furthermore, the extended research model adds the researchers' basic of knowledge on technology acceptance and will help to conduct a wider range of research.

## 2. Theoretical Background and Hypothesis Development

### 2.1. Theoretical Background

#### 2.1.1. The UTAUT2 Model

The early research related to technology acceptance suggests a technology acceptance model (TAM) using the perspectives of PEOU (Perceived ease of use) and PU (Perceived usefulness) to understand user behavioral intentions [8]. Venkatesh et al. [9] then integrated eight theoretical models to explain the relationship between consumer behavioral intention and user behavior; a theory of reasoned action (TRA), a technology acceptance model (TAM), a motivational model (MM), a theory of planned behavior (TPB), a combined TAM and TPB (C-TAM-TPB), a model for PC utilization (MPCU), innovation diffusion theory (IDT) and social cognitive theory (SCT). They established the Unified Theory of Acceptance and Use of Technology (UTAUT), which extends the existing previous research by presenting four key determinants, namely, performance expectation (PE), effort expectation (EE), social influence (SI), and facilitating conditions (FC). UTAUT has been used extensively to describe an individual's adoption of technology including four moderators (i.e., gender, age, experience, and voluntariness) [25].

However, UTAUT was primarily used to explain why users adopt a technology in organizations [24]. Consequently, Venkatesh et al. [10] systematized the extended UTAUT2 to better explain the perspective of individual consumer technology acceptance rather than using the UTAUT model by proposing seven factors and adding three cognitive variables, i.e., hedonic motivation (HM), price value (PV), and habit (HA). The UTAUT2 model has been verified the importance of variables through the theoretical review or empirical analysis in various fields [4,13,21,24].

#### 2.1.2. Trust

Trust can be explained as a contextual expectation or belief that decides behavior [16]. Trust is the consumer's subjective belief that the seller will sincerely fulfill its trading obligations [17,26]. Building trust between sellers and customers is the driving force that will excite customer emotions, and these beliefs lead to positive behavior.

Although the innovation of new technology provides greater convenience and various advantages over existing services, the importance of building trust with customers and systems for customer technology acceptance should be emphasized [22,27]. Since the consumer must provide personal information when using the system after adopting the transaction service, there is a need for a trust relationship with the consumer, where SST does not misuse the information [2]. In addition, because consumers are in an inferior position in an online transaction process, when consumers perceive the seller's competitiveness and benevolence, combined with their own satisfaction based on previous transactions, then they can judge the seller's credibility and determine their personal buying behavior [28]. Thus, trust overcomes a consumer's perceptions of uncertainty and risk and helps the consumer engage in trust-related behaviors with web-based vendors, including sharing personal information and/or purchasing [29]. Consequently, trust is a very important variable for purchasing products/services dealing with uncertainty, especially technology-based products [19], and should be emphasized in any model.

### 2.2. Hypothesis Development

#### 2.2.1. The UTAUT2 Model

To assess the kiosk's factors involved with the QSR's customer behavior, the framework of this study applies a determinant proposed by Venkatesh et al. [10].

Performance expectancy was defined to the extent that an individual believes that using this system will help them achieve performance and productivity [30]. Performance expectancy was found

to be an important predicting factor for consumer's behavior toward new technologies [23,31–37]. They explain that if customers perceive more efficiencies from using the system, they are more likely to lead to positive behavioral intentions in using the system. Additionally, in the food service industry, it was assumed that if customers recognize the benefits of self-service technology, their kiosk behavioral intentions will consistently rise [4]. Therefore, it is assumed that the QSR's kiosk can be a key driving force for technology acceptance by providing practical advantages through a convenient using and transaction method.

Effort expectancy is the degree to which we believe that the system will not be difficult to use [38]. Service convenience is started with a customer's fundamental desire to reduce their time and effort [7]. Studies agree with the opinion that if technology is perceived as easier to use, it is more likely to trigger system use behavior [4,12,39]. Han et al. [14] showed that the perceived ease of use has a positive influence on behavioral intention toward kiosks at fast food restaurants. Baba et al. [4] explain that the self-ordering kiosk screen in a quick-service restaurant is not only easy literacy for customers, but also needs to recognize and apply the user's technical understanding degree. The technological advancement aims to provide more convenience to humans. Therefore, it is assumed that if the QSR's kiosk is recognized as easy to use and convenient, acceptance behavior of technology can be activated.

Social influence is the degree to which consumers believe that other people who are important to them (such as family and friends) should use certain skills [10]. When people discover the positive effects of the use of technology by co-workers and those around them, that view starts with the belief that one can gain similar benefits and values as others get by using the same technology at the same time [31]. Social influence has been demonstrated to create a positive relationship between the technology acceptance of mobile food delivery apps [38] and the usage behavior of smartphone diet apps [12]. Consumers who are not experienced with a specific product or service generally rely on WoM (Word of Mouth) for information acquisition [4]. We can actively receive the information not only from family and friends, but also through various means (e.g., SNS, blogger, Twitter, etc.) with the advancement of technology. Therefore, it is assumed that acquiring information of the kiosks by social influence will have a close relationship with the behavior intention.

Facilitating conditions is the degree of personal belief that an organizational and technical infrastructure exists to support the use of the system [9] and the consumers are paying attention to the existence of facilities, resources, and technology infra needed to use the system [32]. Quick service restaurants can contribute to the greater use of kiosks through a variety of promotions [4]. Indeed, mBankig's different methods for promotion and support from organizations can remove barriers to the use of technology and affect that technology adoption [40]. In the context of kiosks, facilitating conditions must include the function to easily access the screen such as composition of system, touch speed, and screen moving. Therefore, it is assumed that QSR's kiosks are more accessible than traditional systems, so they make for an easier to facilitate technology adoption.

Hedonic motivation refers to the degree to which a person's perception of fun or pleasure is perceived from the use of the technology. The tendency to use technology is closely related to the emotional component that is essential pleasure and fun [41]. Being pleased and excited about use of a new high-tech device has a positive impact on the consumer's technology acceptance attitude [42]. Previous studies have shown that hedonic motivations are important key constructs of the UTAUT2 in describing behavioral intention to technology adoption [35,36,43,44]. Especially, it is emphasized that the perceived fun in using a technology-based self-service has a positive influence on attitudes [45]. The menu touch screen of kiosks make customers participate directly in purchasing behavior. Therefore, it is assumed that the hedonic motivation shown on the kiosk screen will be the key motivation to accelerate the customer behavioral intention.

The perception of value directly affects consumer's willingness to buy [46]. When it is perceived that the benefits of using a technology outweigh the monetary cost, then price value has a positive direction and effect on behavioral intention [10,47]. In terms of price value, if the perceived benefit is high relative to the financial cost being paid, then the customer may be more likely to use a mobile

banking system [31,32]. If the perceived product price by consumers increases beyond the acceptable range, the perception of value also will decrease [46]. With the kiosk's limited service offering, customers will perceive the price value more importantly. Therefore, it is assumed that price value is an essential variable for technology acceptance behavior.

Lastly, habit is an acquired behavior pattern that urges the need to use the self-ordering kiosks [4]. Consumers can develop habits of varying levels, depending on their degree of interaction and familiarity with the developed technology over time [24]. Palau-Saumell et al. [13] confirmed that the user habits in accepting mobile apps for restaurants is the strongest predictor of intention to use and actual use. It was verified that the behavioral intention of customers to order food by using food delivery apps depends on their habits [38]. Consumers' mobile banking habits were found to be the most important antecedent factor affecting behavioral intentions [35]. Therefore, it is assumed that the kiosk's using habit will trigger the continuous behavior in QSR.

As a result, this study considers the key factors of seven (e.g., PE, EE, SI, FC, HM, PV, and HA) about a kiosk characteristic by the comprehensive reviews of the previous study. Many of the preceding studies have suggested that the seven factors of the UTAUT2 are the best source for positive behavioral intentions. The following seven hypotheses were thus established:

**Hypothesis 1 (H1).** *Performance expectancy positively relates to behavioral intentions.*

**Hypothesis 2 (H2).** *Effort expectancy positively relates to behavioral intentions.*

**Hypothesis 3 (H3).** *Social influence positively relates to behavioral intentions.*

**Hypothesis 4 (H4).** *Facilitated conditions positively relate to behavioral intentions.*

**Hypothesis 5 (H5).** *Hedonic motivation positively relates to behavioral intentions.*

**Hypothesis 6 (H6).** *Price value positively relates to behavioral intentions.*

**Hypothesis 7 (H7).** *Habit positively relate to behavioral intentions.*

Previous studies have focused on understanding the direct influence relationship between the preceding variables of UTAUT2 and behavioral intentions. However, this study tries to examine influence relationship among some variables.

Since the UTAUT and UTAUT2 models were developed based on TAM, which is an early technology acceptance model, perceived ease of use and perceived usefulness in TAM are explained by effort expectancy and performance expectancy in UTAUT and UTAUT2. Many previous studies have suggested that perceived ease of use has a significant influence on perceived usefulness in the technology acceptance model (TAM) [48–50]. It was found that the effort expectancy to accept the kiosk technology greatly affects its performance expectancy in the hotel industry [2]. This association means that if the consumer perceives that it is easy to use and there is thus less effort required to use the technology, the system can be used more actively and usefully. On the other hand, since effort expectancy is a personal evaluation and social influences are social pressures, indeed, the effects of social influences have been proven to relate more to performance expectancy rather than to effort expectancy [11]. Researchers have also found that perceived pleasure is an important independent variable of perceived usefulness and ease of use [39,51,52]. In addition, consumers who are more familiar with a website based on previous visits are perceived to believe it is easier to use the site [53]. Consequently, as habit makes it easier to understand, operate, and navigate equipment (e.g., mobile learning platforms), it offers an additional insight into the nature of the positive relationship between habit and effort expectancy [54]. The following five hypotheses were thus established:

**Hypothesis 8 (H8).** *Effort expectancy positively relates to performance expectancy.*

**Hypothesis 9 (H9).** *Social Influence positively relates to performance expectancy.*

**Hypothesis 10 (H10).** *Hedonic motivation positively relates to performance expectancy.*

**Hypothesis 11 (H11).** *Hedonic motivation positively relates to effort expectancy.*

**Hypothesis 12 (H12).** *Habit positively relates to effort expectancy.*

2.2.2. UTAUT2 and Trust

Above all, when consumers properly use the Internet technology infrastructure and Internet-enabled devices, they tend to trust the e-commerce system [47]. Thus, in order to build the trust of the user, a use circumstance with the facilitating conditions is very important. In addition, it was confirmed that the value perceived by consumers represents a strong relationship with trust [55], and furthermore that perceived product value has a positive relationship with students' trust in electronic devices [56]. The following two hypotheses were thus established:

**Hypothesis 13 (H13).** *Facilitating conditions positively relate to trust.*

**Hypothesis 14 (H14).** *Price value positively relates to trust.*

Trust is a major variable for improving the performance expectancy of information communication technology (ICT) services [27] and Internet banking services [37]. Trust in the Internet enables the advantageous expectation that users can rely on and predict the Internet, and that no harmful effects will occur when online consumers use the Internet as a medium for their financial transactions [18]. In addition, trust in acceptance of mobile payment (MP) technology based on near-field communication (NFC) on restaurants was found to have a positive effect on effort expectancy [11]. Building trust in an e-commerce environment also plays an important role in solidifying a customer's willingness to make purchase decisions. Trust reduces uncertainty by creating positive attitudes and behavioral controls for ongoing transactions with web retailers, and trust has a positive impact on consumer behavioral intentions by providing satisfactory expectations for those transactions [20]. Additionally, perceived trust was presented as a predictor on restaurant use intention [13] and Internet banking behavior [26,33,57]. The following three hypotheses were thus established.

**Hypothesis 15 (H15).** *Trust positively relates to performance expectancy.*

**Hypothesis 16 (H16).** *Trust positively relates to effort expectancy.*

**Hypothesis 17 (H17).** *Trust positively relates to behavioral intention.*

*2.3. The Research Model*

This study presents fundamental data for marketing strategy establishment for QSR kiosk technology acceptance based on the proposed hypotheses. Above all, this study investigates the relationship between the seven factors and behavioral intention in the basic UTAUT2 model (*Hypothesis 1–7*), and the relationship between the factors of UTAUT2 as further examined (*Hypothesis 8–12*). In addition, this study investigates the influence between trust and its related variables in the UTAUT2 model (*Hypothesis 13–17*). As a result, Figure 1 shows an extended and integrated UTAUT2 to use for verifying the relationship between the basic UTAUT2 model and trust.

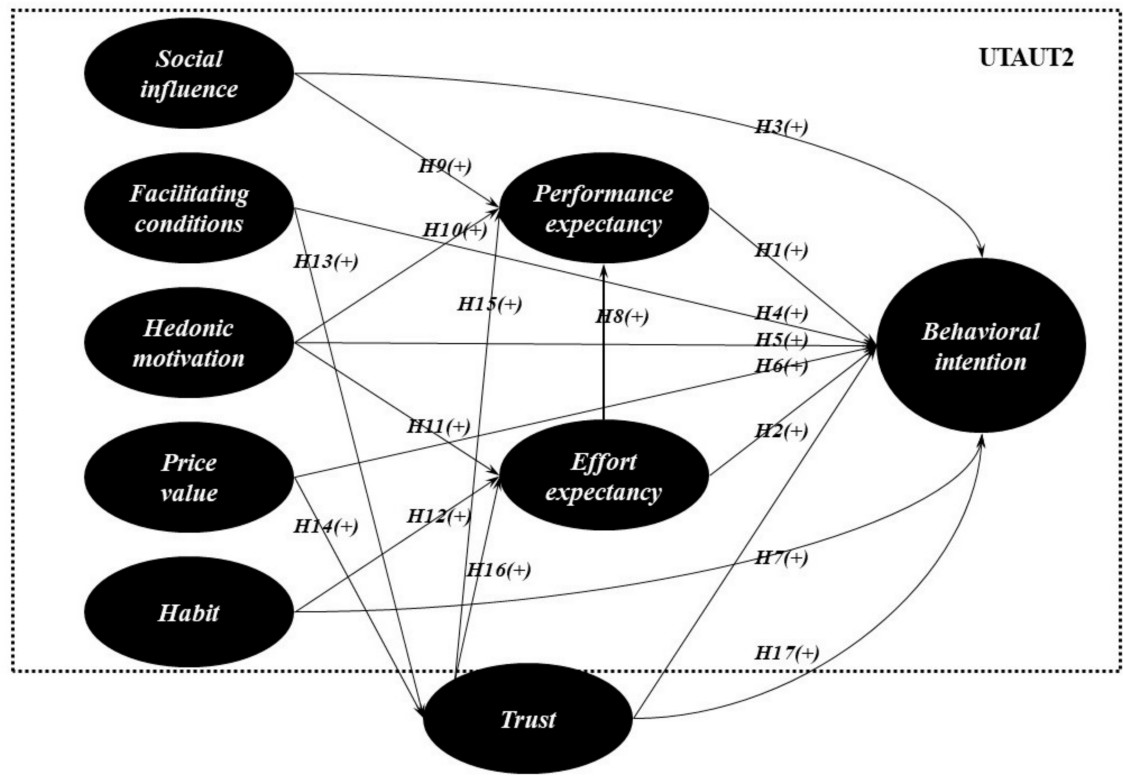

**Figure 1.** A proposed model of UTAUT2 (Unified Theory of Acceptance and Use of Technology) with trust.

## 3. Research Methodology

### 3.1. Samples and Procedures

The sample in this study consisted of customers who were using kiosks in Korean QSR. Prior to this survey being conducted, a preliminary survey was tested on university students who had experience using kiosks in restaurants for 10 days with 50 copies. Any evaluation items that were difficult to understand or had problems were modified. The main survey was distributed from 1 to 20 December 2019, to 350 customers residing in Seoul, Ulsan, Chungcheongbuk-do, and Gyeongsangbuk-do. Prior to any evaluation of this survey, the research purpose and its evaluation method were explained in advance and then conducted after obtaining participant consent, and the data collected from all respondents were kept confidential. Out of 350 surveys, only 303 questionnaires (response rate: 86.6%) were used for the analysis due to missing values and biased responses to the survey.

### 3.2. Instrument Development

The items used to measure for this study hypothesis were evaluated on a 5-point Likert scale (5: strongly agree, 1: strongly disagree). In addition, five items, including gender, age, final education, number of kiosks, and occupation, were additionally evaluated.

#### 3.2.1. UTAUT2 Model

In the UTAUT2, the preceding factors that affected behavioral intention were defined as "a necessary component to determine the kiosk technology acceptance by restaurant visiting customers". The questionnaires of this study were composed based on proposed the UTAUT2 model by Venkatesh et al. [10]. They were divided mainly into seven proven factors that included performance expectancy, effort expectancy, social influence, facilitating conditions, hedonic motivation, price value, and habit items based on the previous research by Venkatesh et al. [10] and Zhou et al. [58], as well as that of

Baptista and Oliveira [35]. For the evaluation, a total of 25 items were used, including "I find kiosks useful in my daily life".

Behavioral intention is defined as "the degree or plans to which the customer intends to visit the restaurant continually through the kiosk's using experience". Based on previous studies [19,27,32], intention was measured using 4 items including "I intend to continue using kiosks in the future".

### 3.2.2. Trust

Trust is defined as a "personal belief that Kiosk's system will protect safely customers' personal information". Based on previous studies [19,23,31], trust was measured using 4 items including "I believe kiosks are trustworthy".

### 3.3. Data Analysis

The collected data was finally analyzed using SPSS (Statistical Package for the Social Sciences) and AMOS (Analysis of Moment Structures). The validity and reliability of the evaluation items were verified using confirmatory factor analysis (CFA) and reliability analysis. In addition, correlation analysis was performed to check the intensity and direction of the variables. It verified seventeen hypotheses using SEM (Structural Equation Model).

## 4. Results

### 4.1. Profiles of the Study Participants

The gender of study participants was 41.6% men and 58.4% women, 2.6% teens, by age group, 60.7% in their 20s, 21.8% in their 30s, and 14.7% in their 40s and over (Table 1). The number of uses of kiosks was 22.4% at 2–4 times/week and 32.0% 1 time/week. Students accounted for the highest percentage of the analysis targets (58.4%).

**Table 1.** Reliabilities and Confirmatory Factor Analysis.

| Construct | Standardized Loadings | t-Value | Composite Reliabilities | Average Variance Extracted | Cronbach's Alpha |
|---|---|---|---|---|---|
| Performance expectancy | | | 0.87 | 0.66 | 0.85 |
| PE1: I find kiosks useful in my daily life. | 0.83 | fixed | | | |
| PE2: I find kiosks efficient in my daily life. | 0.84 | 16.32 *** | | | |
| PE3: Using kiosk services helps me accomplish things more quickly. | 0.77 | 14.65 *** | | | |
| Effort expectancy | | | 0.93 | 0.75 | 0.89 |
| EE1: I find the kiosk easy to use. | 0.79 | fixed | | | |
| EE2: My interaction with the kiosk is clear and understandable. | 0.89 | 17.43 *** | | | |
| EE3: It is easy for me to become skillful at using the kiosk. | 0.92 | 18.00 *** | | | |
| Social influence | | | 0.93 | 0.84 | 0.94 |
| SI1: People who are important to me think that I should use the kiosk. | 0.90 | Fixed | | | |
| SI2: People who influence my behavior think that I should use the kiosk. | 0.95 | 26.47 *** | | | |
| SI3: People whose opinions that I value prefer that I use the kiosk. | 0.91 | 24.29 *** | | | |
| Facilitating conditions | | | 0.85 | 0.68 | 0.81 |
| FC1: I have the knowledge necessary to use the kiosk. | 0.79 | fixed | | | |

**Table 1.** *Cont.*

| Construct | Standardized Loadings | t-Value | Composite Reliabilities | Average Variance Extracted | Cronbach's Alpha |
|---|---|---|---|---|---|
| FC2: The restaurant has the resources necessary to use the kiosk, so I am convenient to use the kiosk. | 0.86 | 15.86 *** | | | |
| Hedonic motivation | | | 0.95 | 0.87 | 0.95 |
| HM1: Using the kiosk is fun. | 0.92 | fixed | | | |
| HM2: Using the kiosk is enjoyable. | 0.98 | 33.16 *** | | | |
| HM3: Using the kiosk is pleasant. | 0.90 | 25.94 *** | | | |
| Price value | | | 0.92 | 0.75 | 0.89 |
| PV1: The kiosk is a good value for the money. | 0.80 | Fixed | | | |
| PV2: The kiosk is reasonably priced. | 0.91 | 18.26 *** | | | |
| PV3: At the current price, the kiosk provides a good value. | 0.87 | 17.52 *** | | | |
| Habit | | | 0.89 | 0.75 | 0.89 |
| HA1: The use of the kiosk has become a habit for me. | 0.81 | fixed | | | |
| HA2: I am addicted to using the kiosk. | 0.91 | 18.71 *** | | | |
| HA3: I must use the kiosk. | 0.87 | 17.73 *** | | | |
| Trust | | | 0.95 | 0.76 | 0.93 |
| TR1: I believe the kiosk is trustworthy. | 0.87 | fixed | | | |
| TR2: I believe the kiosk keep its promise. | 0.92 | 22.26 *** | | | |
| TR3: I believe the kiosks do their job correctly. | 0.87 | 20.19 *** | | | |
| TR4: I believe the kiosk is honest. | 0.83 | 18.62 *** | | | |
| Behavioral intention | | | 0.94 | 0.71 | 0.91 |
| BI1: I intend to continue using the kiosk in the future. | 0.83 | Fixed | | | |
| BI2: I will always try to use kiosks in my daily life. | 0.80 | 16.27 *** | | | |
| BI3: I plan to continue to use the kiosk frequently. | 0.89 | 18.80 *** | | | |
| BI4: I predict I would use mobile banking in the future. | 0.86 | 18.01 *** | | | |

Note: $\chi^2$ = 682.08, df = 314, $\chi^2$/df = 2.17, Goodness of Fit Index (GFI) = 0.86, Normed Fit Index (NFI) = 0.92, Comparative Fit Index (CFI) = 0.95, Root Mean Square Error of Approximation (RMSEA) = 0.06; *** $p < 0.001$.

## 4.2. Measurement of Study Reliability and Validity

This study was verified for the validity and reliability of measurement items to verify the proposed hypotheses and theoretical model. A confirmatory factor analysis (CFA) was implemented to determine whether the measured variables reflected the hypothesized latent variables of the multiple-item scales. CCR estimates, ranging from 0.85 to 0.95 above the recommended cut-off of 0.70 [59] were considered acceptable. AVE had to be greater than the 0.50 cut-off for all proposed constructs [59,60]. The results of this study met these requirements (0.66~0.87). Additionally, as shown in Table 2, the discriminant validity of the measurement model was assessed by comparing the AVE values with the squared correlation between constructs [61]. As a result, the validity of the discriminant concept was almost secured. The fit degree of the model was also acceptable ($\chi^2$ = 682.08 ($p < 0.001$); $\chi^2$/df = 2.17, GFI = 0.86, NFI = 0.92, CFI = 0.95, RMSEA = 0.06, RMR = 0.04), as shown in Table 3.

**Table 2.** Correlations Among the Latent Constructs.

| Construct | Discriminant Validity | | | | | | | | |
|---|---|---|---|---|---|---|---|---|---|
| | **1** | **2** | **3** | **4** | **5** | **6** | **7** | **8** | **9** |
| 1. PE | 0.66 [a] | 0.62 [b] | 0.18 | 00.62 | 0.24 | 0.28 | 0.25 | 0.18 | 0.43 |
| 2. EE | | 0.75 | 0.13 | 0.75 | 0.27 | 0.26 | 0.28 | 0.29 | 0.37 |
| 3. SI | | | 0.84 | 0.20 | 0.32 | 0.33 | 0.42 | 0.05 | 0.21 |
| 4. FC | | | | 0.68 | 0.35 | 0.37 | 0.35 | 0.31 | 0.49 |
| 5. HM | | | | | 0.87 | 0.44 | 0.41 | 0.13 | 0.32 |
| 6. PV | | | | | | 0.75 | 0.63 | 0.26 | 0.40 |
| 7. HA | | | | | | | 0.75 | 0.18 | 0.35 |
| 8. TR | | | | | | | | 0.76 | 0.37 |
| 9. BI | | | | | | | | | 0.71 |

Note: [a] Average variance extracted (AVE); [b] matrix entries are the square correlations; performance expectancy (PE), effort expectancy (EE), social influence (SI), facilitating conditions (FC), hedonic motivation (HM), price value (PV), habit (HA), trust (TR), behavioral intention (BI).

**Table 3.** Structural Parameter Estimate.

| Hypothesized Path (Stated as Alternative Hypothesis) | Standardized Path Coefficients | t-Value | Results |
|---|---|---|---|
| H1: PE → BI | 0.31 | 3.47 *** | Supported |
| H2: EE → BI | −0.110. | −1.12 | Rejected |
| H3: SI → BI | 0.05 | 0.75 | Rejected |
| H4: FC → BI | 0.19 | 2.62 ** | Supported |
| H5: HM → BI | 0.10 | 1.58 | Rejected |
| H6: PV → BI | 0.09 | 1.07 | Rejected |
| H7: HA → BI | 0.09 | 0.94 | Rejected |
| H8: EE → PE | 0.72 | 9.51 *** | Supported |
| H9: SI → PE | 0.15 | 2.68 ** | Supported |
| H10: HM → PE | 0.03 | 0.61 | Rejected |
| H11: HM → EE | 0.24 | 3.56 *** | Supported |
| H12: HA → EE | 0.23 | 3.24 ** | Supported |
| H13: FC → TR | 0.31 | 4.21 *** | Supported |
| H14: PV → TR | 0.24 | 3.62 *** | Supported |
| H15: TR → PE | 0.00 | 1.00 | Rejected |
| H16: TR → EE | 0.37 | 6.20 *** | Supported |
| H17: TR → BI | 0.30 | 4.82 *** | Supported |
| Goodness-of-fit statistics | $\chi^2 = 827.08$ ($p < 0.001$) df = 322 $\chi^2$/df = 2.57 GFI = 0.84 NFI = 0.90 CFI = 0.93 RMSEA = 0.07 RMR = 0.06 | | |

Note: ** $p < 0.01$; *** $p < 0.001$; GFI = Goodness of Fit Index; NFI = Normed Fit Index; CFI = Comparative Fit Index; RMSEA = Root Mean Square Error of Approximation, RMR = Root Mean Square Residual; performance expectancy (PE), effort expectancy (EE), social influence (SI), facilitating conditions (FC), hedonic motivation (HM), price value (PV), habit (HA), trust (TR), behavioral intention (BI).

*4.3. SEM*

SEM was used to evaluate the proposed structural model associated with the 17 offered hypotheses. Table 3 presents the results for the estimated model, illustrating the direction and magnitude of the impacts of the standardized path coefficients. As a result of this analysis, the goodness-of-fit index for the final model was $\chi^2 = 827.08$ ($p < 0.001$), $\chi^2$/df = 2.57, GFI = 0.84, NFI = 0.90, CFI = 0.93, RMSEA = 0.07, RMR = 0.06. The final model was found to have an acceptable level of good fit.

Hypothesis (H1, H4) was verified, as a positive relationship between performance expectancy (β = 0.31; *p* < 0.001), facilitating conditions (β = 0.19; *p* < 0.01), and behavioral intention was supported. Hypothesis (H8) was verified, as a positive relationship between effort expectancy and performance expectancy (β = 0.72; *p* < 0.001) was supported. Hypothesis (H9) was verified in that a positive relationship between social influence and performance expectancy (β = 0.15; *p* < 0.01) was supported. Hypothesis (H11) was proven, as a positive relationship between hedonic motivation and effort expectancy (β = 0.24; *p* < 0.001) was supported. Hypothesis (H12) was verified, as a positive relationship between habit and effort expectancy (β = 0.23; *p* < 0.01) was supported. Hypothesis (H13, H14) was proven as a positive relationship between facilitating conditions (β = 0.31; *p* < 0.001), price value (β = 0.24; *p* < 0.001), and trust was supported. Hypothesis (H16, H17) was verified, as a positive relationship between trust, effort expectancy (β = 0.37; *p* < 0.001), and behavioral intention (β = 0.30; *p* < 0.001) was supported. As a result, the proposed 17 hypotheses were partially adopted.

## 5. Discussion and Implications of the Research

### 5.1. Discussion of Results

Managers need to understand the customer's technology acceptance behavior for kiosks (unmanned payment systems), as kiosks are being actively introduced in many quick service restaurants. Finding the major variables that relate to technology acceptance is of paramount importance when suggesting the direction of the food service industry in the future. The results of this study are expected to provide fundamental data on the efficient operation of restaurant kiosks and new marketing strategies based on the consumer's point of view in this fiercely competitive environment. Furthermore, this effort contributes continuous academic development by presenting the future research direction for management performance improvement.

First, it was confirmed that performance expectancy and a facilitating condition directly affect the behavioral intention or the preceding variables of the technology acceptance model. This result is consistent with the result that the perceived usefulness positively influenced the customer behavior in fast food restaurants' kiosks [14]. In addition, it supports the hypothesis that facilitating conditions in using of self-ordering kiosks have a positive influence on customer purchasing behavior in quick service restaurant [4]. However, the other presented variables were not directly affected by behavioral intention, unlike previous studies. Nevertheless, this study is consistent with the previous studies' results where effort expectancy [35], social influence [32,37,40], hedonic motivation [47], price value [35], and habit [44,62] were found to not directly affect behavioral intention. This study shows that, first of all, the QSR which is low in loyalty is judged to be different from the existing technology acceptance of internet or mobile banking. Next, the simple system of the kiosk is easy for customers to use, and shows that a lack of variety can give pleasure and fun to consumers. Since kiosks are frequently used in most of QSR due to the fast introduction of technology, it is judged that social influences, price values, and habits have no direct effect on behavioral intentions.

Additionally, it was confirmed that as ease of use of a technology increases, the performance expectancy for how that technology use can be perceived is more positive, similar to the results of previous studies [17]. The reason is that if customers perceive that a kiosk is easy or convenient to use, they feel a stronger individual view about the efficiency of using technology. Social influence did not have a significant direct effect on behavioral intention like the results presented earlier, but it did directly affect performance expectancy. In other words, it confirmed the remarkable result that if you do not recognize the efficiency of the kiosk, you will not use it. On the other hand, perceived pleasure played an important role in the technology acceptance model [39] and supported the results of this study. In addition, this study confirmed that the habits formed through experience with a kiosk can result in more easily recognizing the use of that device.

Second, it was found that facilitating conditions and price value have a significant positive effect on trust, which was consistent with the previous research results [47,55]. If managers provide the

necessary information and knowledge to a customer, then the customer will be more likely to increase their trust recognition through the benefits recognized after using the kiosk technology. In addition, this study confirmed that a reasonable price has the likelihood of further enhancing the trust in the kiosk.

Third, trust appears to have a direct effect on performance expectancy and behavioral intentions, and this result was the same as that of previous studies on technology acceptance [13,22,27]. In particular, trust has the greatest explanation power among the variables that have a positive effect on behavioral intentions. Accordingly, in the uncertain environment that can occur in kiosk e-commerce, the customer's belief in the system is paramount for technology acceptance.

## 5.2. Theoretical Implications of Study

The results of this study have very important meanings from an academic point of view.

First, this study has contributed considerably to the existing research by verifying the relationship between the key factors that predict the adoption of kiosk technology in the food service industry. It also goes beyond the scope of previous studies on mobile banking and mobile app technology acceptance. In other words, the UTAUT2 is a materialized theory used to explain technology acceptance from the customer's point of view [10,32], and it was clearly necessary to expand the scope of that research into analyzing QSR.

Second, this study not only verified the relationship between performance expectancy and those facilitating conditions directly affecting behavioral intention, but also effort expectancy, social influence, hedonic motivation, price value, and habit which have an indirect effect on behavioral intention through other related variables. This study identified that all the suggested variables are important in UTAUT2, and the results then suggested a model that is more specific than other studies have offered.

Third, the empirical research in TAM suggests that perceived ease of use is an important variable that directly affects behavioral intention, but in UTAUT and UTAUT2, which are developed based on TAM, the relationship between effort expectancy and behavioral intention was clearly not revealed [30,35,38]. However, this study additionally suggests that effort expectancy has an indirect effect on behavioral intention through the relationship between effort expectancy and performance expectancy in the expanded UTAUT2 model.

Lastly, trust has demonstrated a new causal relationship by directly or indirectly affecting various variables in UTAUT2. This result answers the question, "Is kiosk trust really important?".

## 5.3. Practical Implications of Study

The results of the current study also provide practical implications.

First, the suggested seven factors of UTAUT2 and trust have a positive direct and indirect effect on behavioral intention. As a result, these become important sources for promoting kiosk behavioral intention and present a positive direction for developing strategies for a QSR's sustainable growth. First of all, because customers highly appreciate the efficient benefits and ease of use of the kiosk system, managers must offer the kiosk's unique technical capabilities. Therefore, technology development is continuously necessary to enable ease of communication during the interaction between the customer and this system. Efforts to develop content for stronger communication can lay the foundation for the spread of positive social impact. Additionally, managers should seriously consider the social impacts customers receive from other customers (e.g., word of mouth, blog comments, reviews, etc.). In particular, the facilitating conditions for using the system appear to be powerful predictors that both directly and indirectly influence actual behavioral intention. In other words, the establishment of an infrastructure for the use of technology is an essential element, so restaurants should constantly support more investment in technology development. It was further emphasized that marketing plans should pay special attention to the contribution of hedonic motivation [33]. Managers should focus on providing special promotions through various channels to motivate customers to use kiosks, and web site development should place importance on the hedonic factors that utilize synesthesia, such as the

visual and auditory senses. The customer's menu selecting and ordering through a tablet PC at the table rather than at the store entrance will provide yet another experience environment. Meanwhile, customers have taken significant interest in the monetary value of kiosk technology acceptance. As a result, customers can save time and effort by non-face-to-face services rather than face-to-face services, and they should feel they are receiving a sufficient price value by using a kiosk.

Second, it is very difficult to form an organic relationship between customers and restaurants during non-face-to-face service. Because continuous interaction with people in an uncertain environment creates complex relationships, it is suggested that trust is the most effective way to reduce such uncertainty [63]. As a result, QSR restaurants must consider the critical role of trust in their customer technology acceptance process. This study confirmed remarkable results among price value, trust, and behavioral intention through kiosks. Yet if trust is not the basis, that value is not transferred to behavioral intention. Therefore, the managers have to build a technology infrastructure so that belief between the customer and the system can be formed in the process of self-ordering and receiving services by the customer, and an additional management plan to resolve technical service failures. In particular, when developing a kiosk's application programs, the protection of customer information should be a top priority.

### 5.4. Limitations and Future Research

This study expanded the integrated technology acceptance model (UTAUT2) from the point of view of consumers, following the introduction of kiosks in Korea, and suggested an efficient development plan for restaurants. Still, there were certain limitations. First of all, the results of biased samples in a limited area have limitations for any generalization. The biased samples analyzed came mainly from the 20s–30s group, so were limited for representing all age groups. In addition, the proposed model for this study did not use the moderating analysis (e.g., sex, age, experience, etc.) that was used in previous studies. The potential reason was that as technology advances, every individual must adopt the technology, so voluntariness as a mediator cannot be easily applied [25]. Further, this study is a beginning study used to verify the theorized relationship for UTAUT2 of QSR's kiosks. Future research needs to be re-examined with diverse respondents from various countries. It can present a variety of results depending on cultural differences between countries, and a comparative survey of perceptions between countries will provided interesting results. If this scope of academic research is expanded by supplementing these problems, it is believed it will be possible to improve the understanding of technology acceptance by customers in the food service industry even further and thus perhaps suggest even better development plans.

**Funding:** This research received no external funding.

**Conflicts of Interest:** The author declares no conflict of interest.

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
