# Peer review of "A Study on the Application of Kiosk Service as the Workplace Flexibility: The Determinants of Expanded Technology Adoption and Trust of Quick Service Restaurant Customers"

_sustainability, doi:10.3390/su12218790_

Round 1

Reviewer 1 Report

Dear Author(s),

Many thanks for the opportunity to review this paper.

This research shows a very valuable data related to the technology acceptance of kiosks in QSR (Quick Service Restaurants). Additionally, the research offers an interesting view on the marketing plans for efficient management performance.

I applaud the paper's involvement in technology and marketing plans.

However, I think that the literature review is incomplete.

Please take these examples as an invitation to improve your current theoretical framework:

Jonas, D. (2010). Empowering project portfolio managers: How management involvement impacts project portfolio management performance. International Journal of Project Management28(8), 818-831.

Kim, S. W. (2006). Effects of supply chain management practices, integration and competition capability on performance. Supply Chain Management: An International Journal.

Vorhies, D. W., & Morgan, N. A. (2003). A configuration theory assessment of marketing organization fit with business strategy and its relationship with marketing performance. Journal of marketing67(1), 100-115.

Well, the point is that you can improve the theory section by both adding in relevant literature and by developing a statement of what you find is missing or poorly addressed in the existing literature in the field. This should then lead to analytical principles that are improving the field in terms of a methodological framework that catch the relations between data related to the technology acceptance of kiosks in QSR (Quick Service Restaurants) and marketing plans for efficient management performance in a more direct/proper way.

The paper clearly express its case and uses a correct expression, grammar, sentence structure and any other aspects regarding language and readability.

Author Response

Summary of Revisions

Manuscripts Title:

A Study on the Application of Kiosk Service as the Workplace Flexibility: The Determinants of Expanded Technology Adoption and Trust of Quick Service Restaurant Customers

Manuscripts ID: Sustainability-958528

Note to the Editor

I would like to thank the editor, associate editor and international panel of reviewers for their thoughtful consideration of our manuscripts. The invitation from the editor-in-chief to revise the paper availed us the opportunity to address the two reviewers’ and editor’s remarks. I have thoroughly studied their comments and recommendations and found them very insightful and constructive; they provided valuable guidelines for improving the paper. Indeed, the comments from international panel of two knowledgeable reviewers collectively raised several important concerns effort has been made to address all comments and suggestions.

The following notes explain how I addressed each of the editor’s and reviewers’ concerns (in the order they were raised in the review). I listed each of editor’s/reviewer’s comments first (in italics) and followed it with our responses and explanation of actions.

The major revisions to the paper included (1) improve your current theoretical framework, (2) improve the theory section by both adding in relevant literature, (3) reconstruct the introduction, (4) compare differences with previous studies according to the presented results to make more explicit recommendations.

These revisions have made significant contributions to much improved readability and organization of the manuscript.

Comments to the Author
Many thanks for the opportunity to review this paper. This research shows a very valuable data related to the technology acceptance of kiosks in QSR (Quick Service Restaurants). Additionally, the research offers an interesting view on the marketing plans for efficient management performance. I applaud the paper's involvement in technology and marketing plans.

I would like to thank the reviewer for their thoughtful consideration of our manuscripts. After reading and addressing the reviewer’s comments, we agree that our original manuscripts needed significant revisions. We greatly appreciate the time taken by the reviewer to point out the specific areas where improvements were needs.

Comment#1: However, I think that the literature review is incomplete. Well, the point is that you can improve the theory section by both adding in relevant literature and by developing a statement of what you find is missing or poorly addressed in the existing literature in the field. This should then lead to analytical principles that are improving the field in terms of a methodological framework that catch the relations between data related to the technology acceptance of kiosks in QSR (Quick Service Restaurants) and marketing plans for efficient management performance in a more direct/proper way.

Thank you for your comment. I modified many parts of the study. In the introduction, the problems of the study were clearly presented, and the contribution and goals of the study were rewritten. Please See on Page 1, line 25 ~ Page 3 line 5.

  • The quick-service restaurant (QSR) industry plays an important role in driving the economy and culture around the world. However, the QSR has a particularly difficult problem when creating customer relationships because it is difficult to develop differentiated identities due to a low level of consumer brand loyalty and its standardized products [1]. Nevertheless, companies must build a strong customer-focused relationship marketing strategy to implement their business strategy in a fiercely competitive market environment. In this regard, the existing approach focuses on marketing strategies for customer satisfaction. Until now, studies have not actively reviewed the association with customer behavioral intentions based on a QSR's information communication technology (ICT).
  • The purchasing behavior is one of the strong indicators of the supplier-consumer relationship. Regarding the debate presented earlier, ICT is an important key strategy for successful business activities, so it is thus very important to identify significant predictors that relate to the technology acceptance behavior of users for new technologies.
  • Furthermore, this study additionally further proposes a theoretical framework for the competitive advantage of restaurants. It confirms the importance of trust within the UTAUT2 model. Customers ask themselves questions before using the new system; 'Do I trust the system?'.
  • This study aims to provide fundamental data on Kiosk's marketing strategy plan leading to customer behavioral intentions to QSR's manager. There is a need first to identify the independent factors involved in customer behavioral intentions based on the UTAUT2 model. In addition, it reviews the notion that perceived trust of kiosks is a key factor for determining customer technology acceptance. By integrating the UTAUT2 model and trust, this study presents an extended UTAUT2 model of QSR. This extended model offer guidance to managers for creating the most effective strategies to gather new customers and retain loyal customers, and these action will help with the successful performance of QSR. Furthermore, the extended research model adds the researchers' basic of knowledge on technology acceptance and will help to conduct a wider range of research.
  •  

In addition, I comprehensively improved the theoretical backgrounds, adding other previous studies. The goal of the revised manuscript was to derive the framework necessary to establish a marketing plan. Please See on Page 4, line 3 ~ Page 5, line 33.

To assessment the Kiosk's factors involved with the QSR's customer behavior, the framework of this study apply a determinant proposed by Venkatesh et al. [10].

Performance expectancy was defined to the extent that an individual believes that using this system will help them achieve performance and productivity [30]. Performance expectancy was found to be an important predicting factor for consumer's behavior toward new technologies [23, 31, 32, 34, 35, 36, 37]. They explain that if customers perceive more efficiencies from using the system, they are more likely to lead to positive behavioral intentions in using the system. Also, in the food service industry, it was assumed that if customers recognize the benefits of self-service technology, their kiosk behavioral intention will consistently rise [4]. Therefore, it is assumed that the QSR's kiosk can be a key driving force for technology acceptance by providing practical advantages through a convenient using and transaction method.

Effort expectancy is the degree to which we believe that the system will not be difficult to use [38]. Service convenience is started with customer's fundamental desire to reduce their time and effort [7]. Studies agree with the opinion that if technology is perceived as easier to use, it is more likely to trigger system use behavior [4, 12, 39]. And Han et al. [14] showed that the perceived ease of use has a positive influence on behavioral intention toward kiosks at fast food restaurants. Baba et al. [4] explains that the self-ordering kiosk screen in a quick-service restaurant is not only easy literacy for customers, but also needs to recognize and apply the user's technical understanding degree. The technological advancement aims to provide more convenience to humans. Therefore, it is assumed that if the QSR's kiosk is recognized as easy to use and convenient, acceptance behavior of technology can be activated.

Social influence is the degree to which consumers believe that other people who are important to them (such as family and friends) should use certain skills [10]. When people discover the positive effects of the use of technology by co-workers and those around them, that view starts with the belief that one can gain similar benefits and values ​​as others get by using the same technology at the same time [31]. Social influence has been demonstrated to create a positive relationship between the technology acceptance of mobile food delivery apps [38] and the usage behavior of Smartphone diet apps [12]. Consumers who have not experienced with a specific product or service, consumers generally rely on WoM for information acquisition [4]. We can actively receive the information not only with family and friends, but also through various means (e. g. SNS, blogger, Twitter, etc.) with the advancement of technology. Therefore, it is assumed that acquiring information of the kiosk by social influence will have a close relationship with the behavior intention.

Facilitating conditions is the degree of personal belief that an organizational and technical infrastructure exists to support the use of the system [9] and the consumers are paying attention to the existence of facilities, resources, and technology infra needed to use the system [32]. Quick service restaurants can contribute to the greater use of kiosks through a variety of promotions [4]. Indeed, mBankig's different methods for promotion and support from organizations can remove barriers to the use of technology and affect that technology adoption [40]. In the context of kiosk, facilitating conditions have to include the function to easily access in the screen such as composition of system, touch speed and screen moving. Therefore, it is assumed that QSR's kiosks are more accessible than traditional systems, so they makes easier to facilitate technology adoption.

Hedonic motivation refers to the degree to which a person perception of fun or pleasure perceived from the use of the technology. The tendency to use technology is closely related to the emotional component that is essential pleasure and fun [41]. Being pleased and excited about used of a new high-tech devices has a positive impact on the consumer's technology acceptance attitude [42]. Previous studies have shown that hedonic motivations is an important key constructs of the UTAUT2 in describing behavioral intention to technology adoption [35, 36, 43, 44]. Especially, it is emphasized that the perceived fun in using a technology-based self-service has a positive influence on attitudes [45]. The menu touch screen of kiosk make participate customers directly in purchasing behavior. Therefore, it is assumed that the hedonic motivation shown on the kiosk screen will be the key motivation to accelerate the customer behavioral intention.

The perception of value directly affects consumer's willingness to buy [46]. When it is perceived that the benefits of using a technology outweigh the monetary cost, then price value has positive direction and effect on behavioral intention [10, 47]. In terms of price value, if the perceived benefit is high relative to the financial cost being paid, then the customer may be more likely to use a mobile banking system [31, 32]. If the perceived product price by consumers increases beyond the acceptable range, the perception of value also will decreases [46]. With Kiosk's limited service offering, customers will perceive the price value more importantly. Therefore, it is assumed that price value is an essential variable for technology acceptance behavior.

Lastly, habit is an acquired behavior patterns that urge the need to use the self-ordering kiosks [4]. Consumers can be developed habits of varying levels, depending on their degree of interaction and familiarity with the developed technology over time [24]. Palau-Saumell et al. [13] confirmed that the user habits in accepting of mobile apps for restaurants is the strongest predictor of intention to use and actual use. And it was verified that the behavioral intention of customers to order food by using Food delivery Apps depends on their habits [38]. Consumers' mobile banking habits were found to be the most important antecedent factor affecting behavioral intentions [35]. Therefore, it is assumed that the Kiosk's using habit will trigger the continuous behavior in QSR.

References added to our paper as a result of the revised manuscript.

Kulviewat, S.; Bruner â…¡, G. C.; Kumar, A.; Nasco, S. a.; Clark, T. Toward a Unified Theory of Consumer Acceptance Technology. Psychology & Marketing, 2007, 24(12), 1059-1084.

Dabholkar, P. A.; Bagozzi, R. P. An Attitudinal Model of Technology-based Self-Service: Moderating Effects of Consumer Traits and Situational Factors. Academy of Marketing Science Journal. 2002, 30, 184-201.

Dodds, W. B.; Monroe, K. B.; Grewal, D. Effects of Price, Brand, and Store Information on Buyer’s Product Evaluations. Journal of Marketing research. 1991, August, 307-319.

Tamilmani, K.; Rana N. P.; Prakasam, N.; Dwivwdi, Y. K. The battle of Brain vs. Heart: A literature review and meta-analysis of "hedonic motivation" use in UTAUT2. International Journal of Information Management. 2019, 46, 222-235.

Reviewer 2 Report

It is an interesting paper that deals with a highly relevant topic such as the technology adoption in the restaurant field. The bibliography is extensive and with many recent articles. Some suggestions for improvement are made:

- I think researchers had better to reconstruct the introduction for a better flow of the text. By explaining why they did this research and their reasons. Here are some main issues to answer for the researcher(s) to construct their intro:

- What is the issue that needs to be addressed or solved?

- What is missing in the literature? How is the study going to contribute to the existing knowledge?

- Who is going to benefit from the new knowledge?

- What do researchers want to accomplish by conducting this study?

- What do researchers want to examine/explore/ understand about the topic?

- The research questions are not well developed. It would be necessary to clarify the objectives and argued. Furthermore, the authors present a long battery of hypotheses. It would be interesting to justify and argue each one of them. Not presenting a global argument.

- The fieldwork is supposed to have been carried out in one country, Korea. It would have been interesting, perhaps as a future line of research, to study the users of quick service restaurants from other countries, to see if the cultural differences can affect the results.

- The methodology used can be deemed appropriate, and the authors’ presentation of the results is clear and concise, thus facilitating the reader’s understanding, but it would be necessary to deepen in the limitations of the chosen methodology. It would be interesting to know whether the results presented differ from previous studies. This would add value to the findings.

- I have to say that I do not have access to the tables or figures so my comments may be limited. Furthermore, it would also be interesting to have the questionnaire. What is the origin of the questions? Why were those items selected and not others?

- At the end of the discussion section it would be interesting deep in managerial implications.

Author Response

Summary of Revisions

Manuscripts Title:

A Study on the Application of Kiosk Service as the Workplace Flexibility: The Determinants of Expanded Technology Adoption and Trust of Quick Service Restaurant Customers

Manuscripts ID: Sustainability-958528

Note to the Editor

I would like to thank the editor, associate editor and international panel of reviewers for their thoughtful consideration of our manuscripts. The invitation from the editor-in-chief to revise the paper availed us the opportunity to address the two reviewers’ and editor’s remarks. I have thoroughly studied their comments and recommendations and found them very insightful and constructive; they provided valuable guidelines for improving the paper. Indeed, the comments from international panel of two knowledgeable reviewers collectively raised several important concerns effort has been made to address all comments and suggestions.

The following notes explain how I addressed each of the editor’s and reviewers’ concerns (in the order they were raised in the review). I listed each of editor’s/reviewer’s comments first (in italics) and followed it with our responses and explanation of actions.

The major revisions to the paper included (1) improve your current theoretical framework, (2) improve the theory section by both adding in relevant literature, (3) reconstruct the introduction, (4) compare differences with previous studies according to the presented results to make more explicit recommendations.

These revisions have made significant contributions to much improved readability and organization of the manuscript.

Comments to the Author
It is an interesting paper that deals with a highly relevant topic such as the technology adoption in the restaurant field. The bibliography is extensive and with many recent articles: Some suggestions for improvement are made

I would like to thank the reviewer for their thoughtful consideration of our manuscripts. After reading and addressing the reviewer’s comments, I agree that our original manuscripts needed significant revisions. I greatly appreciate the time taken by the reviewer to point out the specific areas where improvements were needs.

Comment#1: I think researchers had better to reconstruct the introduction for a better flow of the text. By explaining why they did this research and their reasons.

I appreciate editors’ specific and helpful suggestion. I have addressed every suggestion and thank you for helping us strengthen our article. I comprehensively reviewed the introduction. I modified it as follows. Please See on Page 1, line 25 ~ Page 3, line 5.

  • What is the issue that needs to be addressed or solved?

The quick-service restaurant (QSR) industry plays an important role in driving the economy and culture around the world. However, the QSR has a particularly difficult problem when creating customer relationships because it is difficult to develop differentiated identities due to a low level of consumer brand loyalty and its standardized products [1]. Nevertheless, companies must build a strong customer-focused relationship marketing strategy to implement their business strategy in a fiercely competitive market environment. In this regard, the existing approach focuses on marketing strategies for customer satisfaction. Until now, studies have not actively reviewed the association with customer behavioral intentions based on a QSR's information communication technology (ICT). Please See on Page 1, line 25 ~ 33.

  • What is missing in the literature? How is the study going to contribute to the existing knowledge?

Furthermore, this study additionally further proposes a theoretical framework for the competitive advantage of restaurants. It confirms the importance of trust within the UTAUT2 model. Customers ask themselves questions before using the new system; 'Do I trust the system?'. Since unmanned payment systems, such as kiosk services, have various anxiety factors related to technical defects and personal information leakage, users always raise belief questions when using the system. People generally use trust and familiarity as their primary mechanisms to reduce social uncertainty whenever rules and conventions are not enough to convince them [16]. Indeed, Kim et al. [17] emphasize that trust is an important strategy when dealing with an uncertain and uncontrollable future. In terms of technology adoption, trust has been identified as an important aspect in previous studies [18, 19, 20]. A kiosk creates interrelationships between people and technology, and thus trust is paramount in such interrelationships. In the UTAUT2 study, trust was further approached to understand only the direct impact relationship on behavioral intentions [13, 21, 22, 23, 24]. Still we do not fully understand a customer's acceptance of technology because a comprehensive review the UTAUT2 including trust has not yet been conducted. Please See on Page 2, line 31 ~ 48.

  • Who is going to benefit from the new knowledge?

This extended model offer guidance to managers for creating the most effective strategies to gather new customers and retain loyal customers, and these action will help with the successful performance of QSR. Furthermore, the extended research model adds the researchers' basic of knowledge on technology acceptance and will help to conduct a wider range of research. Please See on Page 3, line 1 ~ line 5.

  • What do researchers want to accomplish by conducting this study?

As a result, if you do not understand the relationship that arise between customers and technology in an offline environment, it is difficult to create a specific strategy method for restaurant management, and management can neither be realized for future technological innovation nor business performance successfully. Please See on Page 2, line 8 ~ line 11.

The purchasing behavior is one of the strong indicators of the supplier-consumer relationship. Regarding the debate presented earlier, ICT is an important key strategy for successful business activities, so it is thus very important to identify significant predictors that relate to the technology acceptance behavior of users for new technologies. Please See on Page 2, line 13 ~ line 17.

For all these reasons, it is necessary to research the topic from various perspectives regarding technology adoption for QSR. Above all, based on the UTAUT2 model, it is necessary to comprehensively review various variables related to the customer's behavioral intention of kiosk technology acceptance. Results will be shown clearly the important role of kiosks so that managers can suggest specific marketing strategy ways from a customer perspective. Please See on Page 2, line 27 ~ line 32.

Furthermore, this study additionally further proposes a theoretical framework for the competitive advantage of restaurants. It confirms the importance of trust within the UTAUT2 model. Customers ask themselves questions before using the new system; 'Do I trust the system?'.. Please See on Page 2, line 33 ~ line 35.

Still we do not fully understand a customer's acceptance of technology because a comprehensive review the UTAUT2 including trust has not yet been conducted. Please See on Page 2, line 45 ~ line 46.

  • What do researchers want to examine/explore/ understand about the topic?

This study aims to provide fundamental data on Kiosk's marketing strategy plan leading to customer behavioral intentions to QSR's manager. There is a need first to identify the independent factors involved in customer behavioral intentions based on the UTAUT2 model. In addition, it reviews the notion that perceived trust of kiosks is a key factor for determining customer technology acceptance. By integrating the UTAUT2 model and trust, this study presents an extended UTAUT2 model of QSR. Please See on Page 2, line 47 ~ Page 3, line 1.

Comment#2: The research questions are not well developed. It would be necessary to clarify the objectives and argued. Furthermore, the authors present a long battery of hypotheses. It would be interesting to justify and argue each one of them. Not presenting a global argument.

→ I sincerely appreciate the reviewer’s critical comment. In addition, I comprehensively improved the theoretical backgrounds, adding other previous studies. The goal of the revised manuscript was to derive the framework necessary to establish a marketing plan. Please See on Page 4, line 3 ~ Page 5, line 20.

  • Performance expectancy was defined to the extent that an individual believes that using this system will help them achieve performance and productivity [30]. Performance expectancy was found to be an important predicting factor for consumer's behavior toward new technologies [23, 31, 32, 34, 35, 36, 37]. They explain that if customers perceive more efficiencies from using the system, they are more likely to lead to positive behavioral intentions in using the system. Also, in the food service industry, it was assumed that if customers recognize the benefits of self-service technology, their kiosk behavioral intention will consistently rise [4]. Therefore, it is assumed that the QSR's kiosk can be a key driving force for technology acceptance by providing practical advantages through a convenient using and transaction method.
  • Effort expectancy is the degree to which we believe that the system will not be difficult to use [38]. Service convenience is started with customer's fundamental desire to reduce their time and effort [7]. Studies agree with the opinion that if technology is perceived as easier to use, it is more likely to trigger system use behavior [4, 12, 39]. And Han et al. [14] showed that the perceived ease of use has a positive influence on behavioral intention toward kiosks at fast food restaurants. Baba et al. [4] explains that the self-ordering kiosk screen in a quick-service restaurant is not only easy literacy for customers, but also needs to recognize and apply the user's technical understanding degree. The technological advancement aims to provide more convenience to humans. Therefore, it is assumed that if the QSR's kiosk is recognized as easy to use and convenient, acceptance behavior of technology can be activated.
  • Social influence is the degree to which consumers believe that other people who are important to them (such as family and friends) should use certain skills [10]. When people discover the positive effects of the use of technology by co-workers and those around them, that view starts with the belief that one can gain similar benefits and values ​​as others get by using the same technology at the same time [31]. Social influence has been demonstrated to create a positive relationship between the technology acceptance of mobile food delivery apps [38] and the usage behavior of Smartphone diet apps [12]. Consumers who have not experienced with a specific product or service, consumers generally rely on WoM for information acquisition [4]. We can actively receive the information not only with family and friends, but also through various means (e. g. SNS, blogger, Twitter, etc.) with the advancement of technology. Therefore, it is assumed that acquiring information of the kiosk by social influence will have a close relationship with the behavior intention.
  • Facilitating conditions is the degree of personal belief that an organizational and technical infrastructure exists to support the use of the system [9] and the consumers are paying attention to the existence of facilities, resources, and technology infra needed to use the system [32]. Quick service restaurants can contribute to the greater use of kiosks through a variety of promotions [4]. Indeed, mBankig's different methods for promotion and support from organizations can remove barriers to the use of technology and affect that technology adoption [40]. In the context of kiosk, facilitating conditions have to include the function to easily access in the screen such as composition of system, touch speed and screen moving. Therefore, it is assumed that QSR's kiosks are more accessible than traditional systems, so they makes easier to facilitate technology adoption.
  • Hedonic motivation refers to the degree to which a person perception of fun or pleasure perceived from the use of the technology. The tendency to use technology is closely related to the emotional component that is essential pleasure and fun [41]. Being pleased and excited about used of a new high-tech devices has a positive impact on the consumer's technology acceptance attitude [42]. Previous studies have shown that hedonic motivations is an important key constructs of the UTAUT2 in describing behavioral intention to technology adoption [35, 36, 43, 44]. Especially, it is emphasized that the perceived fun in using a technology-based self-service has a positive influence on attitudes [45]. The menu touch screen of kiosk make participate customers directly in purchasing behavior. Therefore, it is assumed that the hedonic motivation shown on the kiosk screen will be the key motivation to accelerate the customer behavioral intention.
  • The perception of value directly affects consumer's willingness to buy [46]. When it is perceived that the benefits of using a technology outweigh the monetary cost, then price value has positive direction and effect on behavioral intention [10, 47]. In terms of price value, if the perceived benefit is high relative to the financial cost being paid, then the customer may be more likely to use a mobile banking system [31, 32]. If the perceived product price by consumers increases beyond the acceptable range, the perception of value also will decreases [46]. With Kiosk's limited service offering, customers will perceive the price value more importantly. Therefore, it is assumed that price value is an essential variable for technology acceptance behavior.
  • Lastly, habit is an acquired behavior patterns that urge the need to use the self-ordering kiosks [4]. Consumers can be developed habits of varying levels, depending on their degree of interaction and familiarity with the developed technology over time [24]. Palau-Saumell et al. [13] confirmed that the user habits in accepting of mobile apps for restaurants is the strongest predictor of intention to use and actual use. And it was verified that the behavioral intention of customers to order food by using Food delivery Apps depends on their habits [38]. Consumers' mobile banking habits were found to be the most important antecedent factor affecting behavioral intentions [35]. Therefore, it is assumed that the Kiosk's using habit will trigger the continuous behavior in QSR.

Comment#3: The fieldwork is supposed to have been carried out in one country, Korea. It would have been interesting, perhaps as a future line of research, to study the users of quick service restaurants from other countries, to see if the cultural differences can affect the results.

I sincerely appreciate the reviewer’s critical comment on questioning the implications of this study. I know not fully elaborate the contributions of this study. I will review it through related research in the future. I modified in Implications and limitations as follows.

  • Future research needs to be re-examine with diverse respondents from various countries. It can be present a variety of results depending on cultural differences between countries, and a comparative survey of perceptions between countries is will provided interesting results. Please see on Page 11, line 5 ~ line 8.

Comment#4: The methodology used can be deemed appropriate, and the authors’ presentation of the results is clear and concise, thus facilitating the reader’s understanding, but it would be necessary to deepen in the limitations of the chosen methodology. It would be interesting to know whether the results presented differ from previous studies. This would add value to the findings.

I would like to thank the reviewer for their thoughtful consideration of our manuscripts. It is presented in comparison with previous studies as follows.

  • First, it was confirmed that performance expectancy and a facilitating condition directly affect behavioral intention or the preceding variables of the technology acceptance model. This result is consistent with the result that the perceived usefulness positively influenced the customer behavioral in fast food restaurants' Kiosks [14]. In addition, it support the hypothesis that facilitating conditions in using of self-ordering kiosks have a positive influence on customer purchasing behavior in quick service restaurant [4]. However, the presented other variables were not directly affected behavioral intention, unlike previous studies. Nevertheless, this study is consistent with the previous studies results where effort expectancy [35], social influence [32, 37, 40], hedonic motivation [47], price value [35], and habit [44, 62] were found not directly affect behavioral intention. This study results, first of all, the QSR where is low in loyalty, is judged to be different from the existing technology acceptance of internet or mobile banking. Next, the simple system of the kiosk is easy for customers to use, and it be shown lacke a variety that can give pleasure and fun to consumers. And since kiosk is frequently used in most of QSR due to the fast introduction of technology, it is judged that social influences, price values, and habits have no direct effect on behavioral intentions. Please see on Page 9, line 2 ~ line 15.

Comment#5: I have to say that I do not have access to the tables or figures so my comments may be limited. Furthermore, it would also be interesting to have the questionnaire. What is the origin of the questions? Why were those items selected and not others?

I thank the reviewer for this comment. The questionnaire questions suggest in Table 1. In addition, the reasons for selecting the questionnaire were written as follows.

  • The questionnaires of this study were composed based on proposed the UTAUT2 model by Venkatesh et al. [10]. Please see on Page 7, line 20 ~ 21.

Comment#6: At the end of the discussion section it would be interesting deep in managerial implications.

Thank you for your comment. This study suggested both theoretical and practical implications. Please See on Page 9, line 36 ~ Page 10, 43.
